# SITUATEDGEN: Incorporating Geographical and Temporal Contexts into Generative Commonsense Reasoning

**Yunxiang Zhang**
University of Michigan
Ann Arbor, USA
`yunxiang@umich.edu`

**Xiaojun Wan**
Peking University
Beijing, China
`wanxiaojun@pku.edu.cn`

## Abstract

Recently, commonsense reasoning in text generation has attracted much attention. Generative commonsense reasoning is the task that requires machines, given a group of keywords, to compose a single coherent sentence with commonsense plausibility. While existing datasets targeting generative commonsense reasoning focus on everyday scenarios, it is unclear how well machines reason under specific geographical and temporal contexts. We formalize this challenging task as SITUATEDGEN, where machines with commonsense should generate a pair of contrastive sentences given a group of keywords including geographical or temporal entities. We introduce a corresponding English dataset consisting of 8,268 contrastive sentence pairs, which are built upon several existing commonsense reasoning benchmarks with minimal manual labor. Experiments show that state-of-the-art generative language models struggle to generate sentences with commonsense plausibility and still lag far behind human performance. Our dataset is publicly available at `https://github.com/yunx-z/situated_gen`.

## 1   Introduction

In recent years, there has been substantial growth in new benchmarks evaluating commonsense reasoning for natural language processing (NLP) models, especially large-scale Pretrained Language Models (PLMs). Most existing commonsense reasoning benchmarks adopt natural language *understanding* formats due to easy evaluation (e.g., accuracy), including multiple-choice question answering [44, 41, 20, 24], natural language inference [4], and detecting true/false statements [33, 43]. However, datasets measuring commonsense knowledge in natural language *generation* are still relatively scarce. We aim to fill this research gap with a novel benchmark since real-world users of NLP systems would expect the generated outputs from LMs to be not only grammatically correct but also adhere to commonsense knowledge.

COMMONGEN [25], a generative commonsense reasoning challenge, has attracted wide attention recently. Given a set of keywords (e.g., `{dog, frisbee, catch, throw}`), the task requires models to compose a plausible sentence describing everyday scenario using all the provided keywords (e.g., "*The dog catches the frisbee when the boy throws it.*"). While COMMONGEN focuses on social and physical commonsense in everyday life, it is unclear how well current commonsense generation models reason with factual knowledge about specific entities, which is referred to as *entity commonsense* [33]. In this work, we mainly consider geographical and temporal entities, as they provide extra-linguistic contexts [52] for commonsense reasoning and appear in a significant proportion of existing commonsense benchmarks (Section 4.2). To the best of our knowledge, we are the first to incorporate these situations into generative commonsense reasoning.

37th Conference on Neural Information Processing Systems (NeurIPS 2023) Track on Datasets and Benchmarks.

Furthermore, we argue that geographical and temporal contexts are important for commonsense reasoning. On the one hand, basic knowledge about geography and time is part of human commonsense [1, 6], such as "*Earth rotates on its axis once in 24 hours.*" On the other hand, certain types of commonsense knowledge are correlated with specific situations [50]. For example, "*July is summer*" is true for people living in the northern hemisphere, while those living in the southern hemisphere would agree that "*July is winter*".

Our proposed task SITUATEDGEN (**Situated Gen**erative Commonsense Reasoning) requires the machines to generate a pair of contrastive sentences (formally speaking, *antithesis*) with commonsense plausibility, given a group of keywords including geographical or temporal entities. For example, when provided with [July, United States, winter, Australia, summer, July], a reasonable output could be "*July is summer in the United States. July is winter in Australia.*", while a slightly different version "*July is summer in Australia. July is winter in the United States.*" does not adhere to commonsense.

The main challenge for machines to solve the SITUATEDGEN task lies in *situated semantic matching*. In order to generate a pair of contrastive sentences, machines need to split the keywords into two groups (either explicitly or implicitly) based on geographical/temporal relevance and perform relational reasoning [31] within/between the keyword groups.

To study the challenging SITUATEDGEN task, we construct a corresponding large-scale English dataset containing 8,268 pairs of situated commonsense statements. We design an automatic pipeline to collect data at scale with quality assurance and minimal human annotation efforts. Concretely, we derive commonsense statements with geographical or temporal contexts from existing commonsense benchmarks and mine contrastive sentence pairs based on entity-masked sentence similarity. We further manually filter out invalid examples in the test set to ensure the evaluation soundness. To assess the difficulty of our dataset, we conduct automatic evaluations on various generative (large) language models, including BART [22], T5 [39], and InstructGPT [34]. Results show these models lag far behind human performance, indicating that current models struggle to generate sentences adhering to commonsense under the SITUATEDGEN setting. We believe that SITUATEDGEN could serve as a complement to COMMONGEN and enrich the resource for evaluating constrained commonsense text generation in a more realistic setting.

The contributions of this work are three-fold:

- **Task.** We incorporate geographical and temporal contexts into generative commonsense reasoning and propose a novel task SITUATEDGEN.

- **Resource.** We construct a large-scale dataset in a non-trivial way to facilitate the studies of situated generative commonsense reasoning. The dataset is released and will contribute to the commonsense reasoning community.

- **Evaluation.** We benchmark the performance of state-of-the-art generative language models on our dataset and demonstrate the difficulty of the task with a significant gap between machine and human performance.

## 2 Related Work

**Constrained Commonsense Text Generation.** Constrained Commonsense Text Generation [5] requires PLMs to generate commonsense text subject to a set of constraints. Commonsense generation models are currently evaluated by three tasks. First, COMMONSENSE EXPLANATION aims to generate an explanation for why a model selects a candidate answer to a given question. Second, $\alpha$ NLG [4] is another commonsense generation task. The artificial intelligence models are provided with two observations in chronological order and need to generate a plausible hypothesis/explanation describing what happened between the observations. Third, in COMMONGEN [25], models should compose a plausible sentence describing everyday scenarios using all the provided concepts. This task has attracted much attention recently, and researchers advance machine performance on the dataset with contrastive learning [23], prototype editing [28], scene knowledge graph [47], etc. Our proposed task differs from these tasks with a focus on composing a *pair* of contrastive sentences instead of a *single* sentence and incorporating extra-linguistic contexts.

**NLP Benchmarks with Geographical and Temporal Contexts.** There are many emerging benchmarks in NLP that incorporate extra-linguistic contexts such as geographical and temporal contexts. TEMPLAMA [15] and GEOMLAMA [49] probe language models with masked text prompts to query geographical and temporal knowledge. In question answering, MCTACO [54], TORQUE [32] and TIMEQA [9] contains challenging questions involving temporal commonsense reasoning over the duration, frequency, temporal order, and other various aspects of events. SITUATEDQA [52] is made up of open-domain questions whose answers vary across different geographical and temporal contexts. TIMEDIAL [38] studies temporal reasoning in dialogues with a multiple-choice cloze task. In vision-and-language tasks, GD-VCR [50] and MaRVL [27] aim to collect commonsense questions and statements that are visually grounded and geographically diverse. Previous work mainly focuses on how well language models trained on a specific snapshot of corpus can adapt to different contexts. While our dataset SITUATEDGEN also considers such geographical and temporal contexts in language, we probe LMs for a new skill of reasoning for the commonsense relationship among extra-linguistic contexts. We also choose a different task format of generative commonsense reasoning, pioneered by [25], as it focuses on the commonsense reasoning capabilities of generative models rather than NLU models, which is under-researched by the community.

## 3 Task Definitions and Challenges

We use antithesis generation for evaluating generative commonsense reasoning under extra-linguistic contexts. In this section, we first introduce the definitions of our proposed task, followed by an analysis of the main challenges.

### 3.1 Definitions

**Antithesis.** Antithesis refers to a figure of speech that expresses an opposition of ideas with a parallel grammatical structure of words, clauses, or sentences [29, 8]. An example of antithesis could be Neil Armstrong's famous quote "*That's one small step for a man, one giant leap for mankind*". In this work, we adopt a narrow sense of sentence-level antithesis, which means that two simple sentences with similar syntactic structures create a contradiction in semantics. Intuitively, the qualifying two sentences can be connected into a coherent sentence via conjunction words such as "while", "yet", and "whereas" (e.g., "*July is summer in the United States, while July is winter in Australia.*"). We emphasize commonsense plausibility rather than the rhetorical effect of antithesis within the scope of this paper.

**Extra-Linguistic Contexts.** Following [52], we focus on two context types: geographical (GEO) and temporal (TEMP). GEO defines each context value as a geopolitical entity ("GPE"). TEMP defines each context value as timestamp ("DATE", "TIME", "EVENT").

**Contextual Dependence.** We define that a contrastive sentence pair is *context-dependent* if swapping any of the GEO or TEMP entities between the two sentences could lead to a contradiction with commonsense yet grammatical correctness. For example, for the sentence pair "*July is summer in China. July is winter in Australia.*", if the two GEO entities "China" and "Australia" are swapped, the resulting sentences do not adhere to commonsense anymore: "*July is summer in Australia. July is winter in China.*" This indicates that they are context-dependent.

Contextual dependence is crucial for a proper evaluation of the generation results. Because sentence pairs that do not satisfy context dependence may have multiple valid answers (swapping the entity words leads to an extra correct answer), the metrics introduced in Section 6 cannot make a sound evaluation with only a single reference.

**Situated Generative Commonsense Reasoning.** We modify the mathematical formulation of the task COMMONGEN to define SITUATEDGEN. The input of the task is a multiset[1] consisting of $k$ keywords $x = [c_1, c_2, ..., c_k] \in \mathcal{X}$, where each keyword $c_i \in \mathcal{C}$ is a noun or entity, a single word or phrase. We denote $\mathcal{X}$ as all possible combinations of keywords and $\mathcal{C}$ as the vocabulary of keywords.

---

[1]Multiset is a set that allows multiple instances for each of its elements.

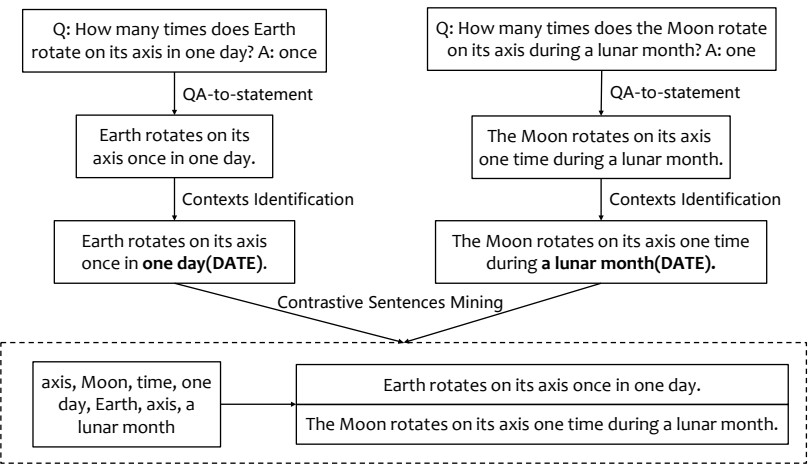

Figure 1: An overview of data collection pipeline. Inside the dotted box is a final example in the dataset.

Keywords in $x$ should contain at least two GEO or TEMP entities of the same type and two other keywords[2].

The output of the task is an unordered pair of coherent and plausible sentences $y = \{s_1, s_2\} \in \mathcal{Y}$ that satisfies the following conditions: 1) the sentence pair includes all keywords in $x$; 2) each sentence has at least one GEO or TEMP keyword; 3) each sentence is geographical-temporal-semantically correct; 4) $s_1$ and $s_2$ form a pair of contrastive sentences, or antithesis; 5) $s_1$ and $s_2$ are context-dependent. The goal of the task is to learn a function $f : \mathcal{X} \rightarrow \mathcal{Y}$ that maps a group of keywords $x$ to a pair of sentences $y$.

## 3.2 Challenges: Situated Semantic Matching

As the goal of our task is to generate a pair of sentences instead of a single sentence, machines need to explicitly or implicitly classify the keywords into two subgroups based on their geographical and temporal semantic relevance, so as to generate one commonsense sentence with each subgroup. For example, given [July, China, winter, Australia, summer, July], the resulting keyword subgroups should be {July, China, summer} and {July, winter, Australia}.

During the process of keyword grouping and matching, machines need to make connections among keyword concepts with relational reasoning [31] over factual knowledge about these nouns and entities, a.k.a. *entity knowledge* [52], such as geographical location, temporal order, physical rules, social customs, etc. The matching process is important since wrong grouping results will lead to generated sentences without commonsense plausibility[3].

We require that the two sentences should have similar syntactic structures and express similar relationships (e.g., "*X lives in Y*"). This is important for securing the difficulty of the task as it prevents models from learning shortcuts [18, 45] to group keywords based on trivial syntactic (e.g., POS tag of the word) and semantic (e.g., two different kinds of relationship) information. For example, if the two sentences have different syntactic structures (e.g. "X lives in Y" and "Z eats W"), then the model could simply put a city name in Y and a food name in W for keyword grouping and ignore the commonsense connection with X/Z. This type of shortcut reduces the task difficulty.

---

[2]We do not explicitly provide the types of keywords in our dataset. The models are expected to infer which keyword is GEO or TEMP if needed.

[3]We note that under certain circumstances, wrong grouping results might produce correct answers via negative sentences. For example, the machine could generate "*July is **not** summer in Australia*" with {July, Australia, summer}. However, we observe that these are rare scenarios in our datasets, so we do not consider their confusing effects in our study.

# 4 Dataset Collection

To study the SITUATEDGEN challenge, we construct a large-scale English dataset. We design a pipeline to collect high-quality data at scale with minimal manual annotation efforts. Figure 1 illustrates the overall pipeline for dataset collection, which consists of three steps:

1. **QA-to-statement.** Converting question-answer pairs of existing commonsense question answering benchmarks into corresponding statements.

2. **Contexts Identification.** Identifying all entities in a statement with a NER tagger and removing those statements without GEO and TEMP entities.

3. **Contrastive Sentences Mining.** Automatically mining contrastive sentence pairs (antithesis) from the remaining commonsense statements based on entity-masked sentence similarity.

## 4.1 QA-to-Statement

Our dataset is composed of commonsense statements, which are simple sentences describing commonsense knowledge, e.g., *"You would find many canals in Venice."* In recent years, numerous commonsense reasoning benchmarks have been proposed and they form a potentially available commonsense knowledge base with high quality and diverse content. Inspired by recent benchmarks that are sourced from existing datasets [52, 37], we aim to extract commonsense statements from these commonsense benchmarks. We assume that the knowledge in these commonsense benchmarks is *actually* commonsense instead of encyclopedic knowledge, though they might not be shared locally in certain groups of people due to a lack of geographical diversity. That being said, we adopt and follow the concept of "commonsense" widely used in existing works.

We conduct a holistic study of commonsense reasoning datasets to date and select five different data sources after considering their size, annotation quality, and reasoning difficulty. They are CREAK [33], StrategyQA [17], CommonsenseQA [44], ARC [11] and OpenbookQA [30], respectively. We briefly introduce the nature of each dataset in Appendix A.1. Since the raw data come in different formats such as multiple-choice questions and Yes/No questions, we apply a specific preprocessing method for each dataset to transform them (i.e., question-answer pairs) into statements. The transformation details are also included in Appendix A.1. In general, we collected 35,997 commonsense statements from the five source datasets (statistics in Table 1).

## 4.2 Contexts Identification

We now filter out commonsense statements without geographical or temporal contexts. Following [52], we identify sentences with extra-linguistic contexts by GEO and TEMP entities. We use FLERT[4] [42], a named entity recognition (NER) model, to extract all entities from a sentence and remove those statements without any GEO ("GPE") or TEMP ("DATE", "TIME", "EVENT") entities.

Table 1 shows that of all the commonsense statements extracted from the five source datasets, 6.6% sentences have GEO contexts and 5.5% have TEMP contexts, which we count as a significant proportion. Finally, we obtain 4,038 (11.2%) commonsense statements with extra-linguistic contexts.

## 4.3 Contrastive Sentences Mining

We aim to automatically mine contrastive sentence pairs from the commonsense statement corpus. Antithesis mining has not been studied in the existing literature, so we propose a pilot algorithm. We observe that after removing keywords from contrastive sentences, the remaining parts are very similar since antithesis sentences have parallel syntactic structures [8]. Based on this observation, we design the antithesis mining algorithm illustrated in Figure 2 consisting of three steps:

1. **Keyword Masking.** We extract all entities and other nouns as keywords in the sentence and replace each keyword with a [UNK] token, telling the pretrained language models to neglect the meaning of these keywords.

---

[4]https://huggingface.co/flair/ner-english-ontonotes-large

Table 1: Statistics of contexts identification results. "Sent" means the commonsense statements collected in Section 4.1. "GEO"/"TEMP" refer to statements with *only* geographical/temporal entities. "GEO & TEMP" refers to statements with *both* geographical and temporal entities. "Valid Sent" means the commonsense statements with GEO or TEMP contexts.

| Dataset | # Sent | # GEO | # TEMP | # GEO & TEMP | # Valid Sent |
|---|---|---|---|---|---|
| CREAK | 5,779 | 868 | 552 | 153 | 1,573 |
| StrategyQA | 4,976 | 501 | 366 | 86 | 953 |
| CommonsenseQA | 10,962 | 487 | 215 | 12 | 714 |
| ARC | 7,787 | 165 | 426 | 52 | 643 |
| OpenbookQA | 6,493 | 31 | 119 | 5 | 155 |
| Total | 35,997 | 2,052 | 1,678 | 308 | 4,038 |

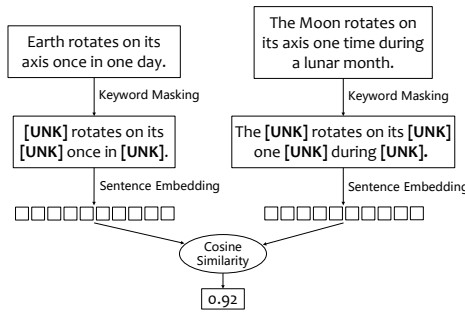

Figure 2: An illustration of the contrastive sentence mining algorithm.

2. **Masked Sentence Similarity Matching.** We obtain the embedding of the keyword-masked sentence from a pretrained language model and calculate the cosine similarity between all possible sentence pairs.

3. **Rule-based Filtering.** We filter out invalid sentence pairs based on a fixed threshold of masked sentence similarity, number of keywords, and entity types.

We introduce the implementation of our antithesis mining algorithm in Appendix A.2. In this way, we efficiently extracted large-scale contrastive sentence pairs from all possible pairwise combinations of the aforementioned commonsense statements with extra-linguistic contexts[5] (Section 4.2). For each contrastive sentence pair, we merge the keywords from each statement and randomly shuffle them to get the input data. The output is the concatenation of two statements.

## 4.4 Dataset Splitting

When splitting the data into training, validation, and test set, we explicitly require that one statement cannot appear simultaneously in any two sets. Consequently, there is no overlap of the single sentence (or sentence-level keyword combinations) among the training, validation, and test data. This requirement forces machines to reason over new combinations of keywords during the inference stage instead of memorizing existing keywords matching results. Statements with similar syntactic structures will also be divided into the same set to reduce overlap of syntactic templates across different sets.

Specifically, we treat dataset splitting as a community structure [7] discovery problem. Community structure refers to a group of tightly connected nodes that have a high density of internal connections and a low density of external connections. We regard a single sentence as a node in the graph. If two single sentences can be matched into a pair of contrastive sentences, an undirected edge will connect the corresponding nodes of these two single sentences. In this way, we obtain an undirected graph describing the dataset structure. A subset of a dataset (such as a training set) is equivalent to a subgraph containing all sentence pairs (edges) and single sentences (nodes) of that subset.

In order to prevent the same sentence from appearing across different sets, we require that the subgraph node sets of the training set, validation set, and test set are disjoint. We use a community structure detection algorithm to meet this requirement. We use the community as the basic unit of dataset splitting, putting all the edges (sentence pairs) in one community into a certain dataset split. Connecting edges between communities (two vertices belonging to different communities) are removed. We note that sentences with similar syntactic structures tend to be connected to each other in the graph and thus fall into the same community, which ensures the syntactic variability between train/dev/test splits.

---

[5]One statement might be paired with multiple statements, formulating multiple contrastive sentence pairs.

Table 2: The basic statistics of the SITUATEDGEN dataset. "Sent" means commonsense statement.

| Statistics | Train | Dev | Test |
|---|---|---|---|
| Size (# Sent Pairs) | 5,641 | 1,407 | 1,220 |
| # Unique Sents | 788 | 309 | 341 |
| per Sent Pair | 0.14 | 0.22 | 0.28 |
| # Unique Keywords | 1,847 | 725 | 851 |
| # Avg. Input Keywords | 7.34 | 6.96 | 6.89 |
| # Avg. Output Tokens | 20.89 | 24.08 | 20.61 |

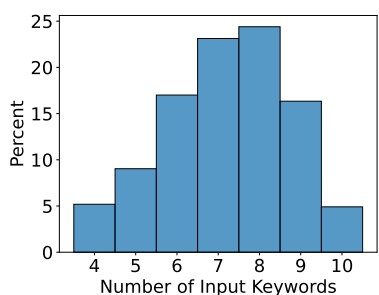

Figure 3: Distribution of numbers of input keywords.

We use the Louvain [7] community structure detection algorithm[6] and divide our graph into 79 communities. The largest community contains 3,273 edges, accounting for about 26% of the total data. We remove edges connecting different communities and then randomly divide the communities of contrastive sentence pairs into training set, validation set or test set.

To ensure the evaluation soundness, we manually filter out invalid examples in the *test* set that are not fluent antitheses or context-dependent. 13.6% of test data is removed and the final dataset has 8,268 examples in total. See additional details of manual filtering in Appendix B.

## 5 Dataset Analysis

### 5.1 Quality Analysis

To measure the quality of our automatically collected data, we randomly select 100 examples (i.e. sentence pairs) from the validation set (which is not manually filtered) and annotate each example for whether it is actually 1) (fluent) antithesis and 2) context-dependent. We find that 87% of the data are real antitheses with fluency and 80% of the data satisfy both of the two requirements. Considering that our dataset is constructed through a fully automatic pipeline, this quality is pretty satisfying and can meet the needs of training and evaluation. As we have discussed in Section 3.1, test examples not satisfying contextual dependence can fool the evaluation metrics, since there are multiple valid references despite the single one provided in the test set. Thanks to the additional manual filtering at the end of Section 4.3, the test set is now qualified for evaluation. As for the unfiltered training set, even if a contrastive sentence pair is not context-dependent, it is still valuable training data, satisfying the other requirements for the target side (Section 3.1). Reduced size of training data after potential manual filtering is also unfavorable to the learning of models. As a result, we retain all the examples in the training set.

Now we analyze the error cases in detail, including non-contrastive and non-context-dependent sentence pairs. The main explanation that accounts for the production of non-contrastive sentence pair is that the remaining verbs after keyword masking may have lexical ambiguity, e.g. "play" in "*Slaves **play** a role in the history of the united states.*" and "*A team sport **played** mostly in Canada is Lacrosse.*" Although the pretrained language models could infer the meaning of a word according to its context [14], the contexts are lost after keyword masking. As a result, two sentences with different syntactic structures are matched together, thus violating the antithesis rule. This poses a limitation of our antithesis mining algorithm.

In addition, 7% of the sentence pairs are antitheses yet not context-dependent. Take the following sentence pair as an example: "*You could find millions of brownstone in New York City.*[7] *One can find a Holiday Inn inside the United States.*". After swapping the GEO entity "New York City" and "United States" in these two sentences, they still conform to commonsense. The reason for this phenomenon is that New York City is part of the United States, and thus the "brownstone" related to New York will also be related to the United States. However, we would like to point out that contextual dependence

---

[6]https://github.com/shobrook/communities
[7]As background knowledge, there are many historical buildings in New York City whose facades are made of brown sandstone, see https://bungalow.com/articles/what-exactly-is-a-brownstone.

is not an absolutely strict condition. Although this example still holds after swapping the GEO entities, it is not the optimal answer, because "brownstone" is more a typical thing in New York City and thus more suitable for a match with "New York City".

## 5.2 Dataset Statistics

Table 2 includes the basic statistics of the SITUATEDGEN dataset. If we use the ratio of unique statement count to sentence pair count ("# Unique Sents per Sent Pair") to represent the content/keyword diversity of the dataset, the validation set, and the test set are relatively high (0.22/0.28), compared to the training set (0.14).

**Distribution of Numbers of Input Keywords.** Figure 3 shows the distribution of numbers of input keywords for all examples in the dataset. Intuitively, more input keywords imply an increased number of possible combinations, making it more difficult for the models to handle. The average number of input keywords is 7.21 and the distribution is fairly symmetrical (skewness=-0.25), suggesting that the SITUATEDGEN has a reasonable difficulty.

**Distribution of Context Types.** Here we define three context types of pairs of contrastive sentences: a GEO pair of sentences contain only GEO entities; a TEMP pair of sentences contain only TEMP entities; If both sentences contain GEO and TEMP entities, the pair of sentences belongs to the type of GEO & TEMP . We find that 78% of all sentence pairs are GEO , 21% are TEMP and the rest 1% are GEO & TEMP .

## 6 Methods

**Baseline Models.** We benchmark the performance of three prominent pretrained language generation models with encoder-decoder architecture — BART [22], T5 [39], FLAN-T5 [10] — and a decoder-only large language model (LLM) — InstructGPT [34] with 175B parameters. We train BART, T5, and FLAN-T5 models in a fully supervised setting with the seq2seq format and expect that the models can learn to group keywords *implicitly*. Specifically, for the input of BART, we concatenate all shuffled keywords with a comma as the separation token "$c_1, c_2, ..., c_k$". Regarding the input format of T5/FLAN-T5, we prepend the keyword sequence with a simple task description to align with its pretraining objective: "*generate two sentences with: $c_1, c_2, ..., c_k$*". The outputs of all models are simple concatenations of the two target sentences $s_1$ and $s_2$. Since the output is an unordered pair, we feed two examples "$x \rightarrow s_1\ s_2$" and "$x \rightarrow s_2\ s_1$" to the model for each original training example. As for InstructGPT, we evaluate it in a few-shot setting. We build prompts with instruction and in-context demonstrations. For each test example, we randomly select 10 training examples as in-context demonstrations. We report the model hyper-parameters and GPT prompt format in Appendix C.1.

**Evaluation Metrics.** [25] have well established the automatic evaluation protocol of the generative commonsense reasoning task. They demonstrated a strong correlation between automatic metrics and human evaluation results. Since SITUATEDGEN adopts a similar format of keyword-to-text generation to COMMONGEN , we follow the evaluation protocol of COMMONGEN and do not include an extra manual evaluation in our study.

Concretely, we employ several widely-used automatic NLG metrics based on n-gram overlap — BLEU [36], ROUGE [26], METEOR [3] — and image caption metrics that focus on the consistency of keywords and their relationships — CIDEr [46] and SPICE [2]. In order to assess the validity of the generated outputs, we include BERTScore [53], a content-oriented and semantic metric. We also adopt COVERAGE, which is the average percentage of input keywords that are present in lemmatized outputs. Additionally, we report the accuracy of keyword grouping results[8] as MATCH, which serves as a good indicator of the commonsense plausibility of the generated texts. See Appendix C.2 for the implementation details of these evaluation metrics.

---

[8]Keywords appearing in the same lemmatized output sentence are considered to be grouped together by models. In particular, if a keyword does not appear in the output, we treat it as unmatched.

Table 3: Experimental results on the test set of SITUATEDGEN . The best model performance is in **bold**. Human performance is tested on a subset of 100 random samples.

| Model (# parameters) | COVERAGE | MATCH | BLEU-4 | ROUGE-2 | METEOR | CIDEr | SPICE | BERTScore |
|---|---|---|---|---|---|---|---|---|
| BART-base (140M) | 78.3 | 60.5 | 22.7 | 29.9 | 29.6 | 18.3 | 53.9 | 48.4 |
| BART-large (400M) | 73.3 | 63.1 | 23.7 | 31.6 | 29.2 | 18.5 | 55.3 | 48.1 |
| T5-base (220M) | 75.6 | 55.3 | 21.9 | 28.7 | 29.8 | 17.4 | 53.6 | 46.2 |
| T5-large (770M) | 81.3 | 67.8 | 26.6 | 33.5 | 31.9 | 21.2 | 57.8 | 51.9 |
| FLAN-T5-base (220M) | 78.0 | 58.7 | 22.3 | 29.5 | 30.6 | 18.2 | 54.7 | 47.6 |
| FLAN-T5-large (770M) | 83.1 | 70.3 | 27.4 | 34.8 | 32.6 | 22.4 | 58.8 | 53.6 |
|    GEO | 83.1 | 70.8 | 26.8 | 33.9 | 32.4 | 21.9 | 58.2 | 52.8 |
|    TEMP | 83.1 | 67.0 | 31.2 | 40.4 | 34.1 | 22.7 | 62.5 | 59.1 |
| InstructGPT (175B, 10-shot) | **91.8** | **79.6** | **28.4** | **36.3** | **36.1** | **23.4** | **60.9** | **56.4** |
| Human | 98.1 | 92.9 | 39.9 | 46.9 | 40.4 | 39.7 | 71.4 | 65.0 |

Table 4: Case studies of machine generations. Keywords appearing in the generation results are underlined.

| | |
|---|---|
| Input Keywords | 24 hours, axis, one month, Earth, axis, Moon |
| Reference | It takes one month for the Moon to rotate on its axis. Earth rotating on its axis takes 24 hours. |
| BART-base | The axis of the Moon is 24 hours. One month is one month. |
| BART-large | There are 24 hours in one month. |
| T5-base | Earth has a 24 hour axis. One month is one month. |
| T5-large | One month is one month on Earth. The Moon is 24 hours away from the axis of the Earth. |
| FLAN-T5-base | The Moon is the axis of the Earth. One month is one month. |
| FLAN-T5-large | The Moon is 24 hours away from Earth. One month is one month. |
| InstructGPT | The Earth takes 24 hours to rotate on its axis. The Moon takes one month to orbit around the Earth. |
| Input Keywords | Paul, Emperor, China, Qin, Russia, dynasty |
| Reference | The Qin dynasty reigned in China. Paul I of Russia reigned as the Emperor of Russia. |
| BART-base | The Emperor of China worked in China. Paul served as the first emperor of the dynasty Qin. |
| BART-large | Emperor of the Qin dynasty. Paul existed in Russia. |
| T5-base | China is a dynasty of China. Paul Qin is the Emperor of China. |
| T5-large | Paul was the Emperor of Russia. The Qin dynasty ruled China. |
| FLAN-T5-base | Paul was the emperor of China. The history of Russia includes the history of Qin. |
| FLAN-T5-large | The Emperor of Russia was Paul the Great. Qin dynasty existed in China and had history in Russia. |
| InstructGPT | Emperor Paul was part of the Russian dynasty. Qin was part of the Chinese dynasty. |

# 7   Results

In Table 3, we report the experimental results of different baseline models on the test set of SITU-ATEDGEN. We approximate human performance with 100 randomly sampled examples from the test set which are annotated by the authors of this paper. We observe that larger models tend to have better performance than smaller ones, as larger parameters store more commonsense knowledge and provide better language generation quality. Notably, the few-shot InstructGPT surpasses other fully-supervised models in every metric, demonstrating its strong reasoning ability. Nevertheless, it still lags far behind human performance. For example, there is a difference of 13.3 points in MATCH, indicating the lack of commonsense in machine generations. The large gap of keyword-oriented metrics (CIDEr and SPICE) also suggests that models find it difficult to infer the relationship between keywords. The significant gap between models and humans demonstrates the difficulty of SITUATEDGEN and leaves much room for improvement in future research.

**Performance across Different Context Types.**   Table 3 reports the performance of the FLAN-T5-large model across different context types. The results show that the matching accuracy of TEMP

type is lower than GEO, indicating that temporal-dependent test examples are more challenging. However, the amount of TEMP data is less than GEO in the training set, which may also give rise to the performance difference. Interestingly, the generation fluency of GEO type is worse than TEMP, suggesting that it is more difficult to use GEO entities to compose sentences smoothly.

**Case Study.** Table 4 shows two groups of generation examples by different models. The first example belongs to TEMP type ("24 hours" and "one month") and the second one is GEO ("Russia" and "China"). We find that models are prone to omit keywords in their outputs. For example, BART-large only covers 2 out of 6 keywords in the first example. Besides, most of the observed generated outputs are not commonsensical due to incorrect keyword grouping results, e.g., "*There are 24 hours in one month*". InstructGPT results seem to have the best generation quality and commonsense plausibility among other models, but it still demonstrates incompetence in handling the contrastive relationships between the two sentences.

# 8   Conclusion

In this paper, we introduce the challenging task SITUATEDGEN to incorporate geographical and temporal contexts into generative commonsense reasoning. We build a corresponding testbed to evaluate the situated reasoning capabilities of state-of-the-art text generation models. The benchmark performance shows that models struggle to generate commonsensical sentences and lag far behind humans. Altogether, our data will serve as a challenging benchmark for measuring commonsense knowledge in generative language models and support research progress of constrained commonsense text generation in a more realistic situation.

## Limitations

1. Since our dataset is derived from existing commonsense benchmarks, we may inherit their annotation artifacts [18] and contain certain types of spurious lexical patterns (e.g., "A lived in B").

2. We do not provide an automatic evaluation of the aspect of contrast between the sentences. A possible solution is to compute the similarity between the entity-masked sentences. This is similar to how we mine contrastive sentences during dataset collection (Figure 2).

3. We could also conduct an extra manual evaluation on the machine generations, so as to gauge its correlation with automatic metrics, though this has been verified by [25] on the original generative commonsense reasoning task.

4. Recently, a lot of work has developed new retrieval-augmented commonsense text generation models [51, 19], which could also be included as baseline models for a more comprehensive benchmark.

## Ethics Statement

Our data is built upon publicly available datasets and we will follow their licenses when releasing our data. There is no explicit detail that leaks an annotator's personal information. The dataset has very low risks of containing sentences with toxicity and offensiveness. Since our data is sourced from existing datasets, we may inherit geographical biases [16] that result in an uneven distribution of commonsense knowledge about western and non-western regions. The commonsense statements may not sound familiar to people who live in locations that are poorly represented in the source datasets. Therefore, models developed on our dataset may preserve biases learned from the annotators of the source datasets. We note that pretrained language models may also inherit the bias in the massive pretraining data. It is important that interested parties carefully address those biases before deploying the model to real-world settings.

## Acknowledgements

We thank the anonymous reviewers for their helpful comments and suggestions.

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

# A Additional Details of Dataset Collection

Table 5: Source dataset examples. **Correct answers** are in bold and underlined.

| Dataset | Size | Format | Raw Data → Statement Conversion Example |
|---|---|---|---|
| CREAK [33] | 13,418 | True/False statement | In the calendar year, May comes after April and before June. (**True**/False) → In the calendar year, May comes after April and before June. |
| StrategyQA [17] | 5,111 | Yes/No Question | Are more watermelons grown in Texas than in Antarctica? (**Yes**/No) → More watermelons are grown in Texas than in Antarctica. |
| CommonsenseQA [44] | 12,247 | Multiple-choice Question | Where in Southern Europe would you find many canals? (A) Michigan (B) New York (C) Amsterdam **(D) Venice** (E) Sydney → You would find many canals in Venice, Southern Europe. |
| ARC [11] | 7,787 | Multiple-choice Question | How long does it take for Earth to rotate on its axis seven times? (A) one day **(B) one week** (C) one month (D) one year → It takes one week for Earth to rotate on its axis seven times. |
| OpenbookQA [30] | 6,493 | Commonsense Statement | You wear shorts in the summer. → You wear shorts in the summer. |

## A.1 Commonsense Statement Collection

We briefly introduce the nature of each source dataset in Section 4.1.

- **CREAK** [33] is a commonsense fact verification dataset featuring entity commonsense, which includes 13,418 true or false statements about entity knowledge written by crowd-workers.

- **StrategyQA** [17] is a commonsense question answering dataset that requires multi-hop implicit reasoning. It consists of 5,111 questions whose answers are either Yes or No. Machines need to decompose a question into multiple atomic questions to arrive at an answer.

- **CommonsenseQA** [44] is a commonsense question answering dataset of 12,247 five-way multiple-choice questions with a focus on knowledge in everyday life.

- **ARC** [11] is a commonsense question answering dataset. It has 7,787 four-way multiple-choice natural science questions collected from grade-school standardized tests.

- **OpenbookQA** [30] is a commonsense question answering dataset that simulates openbook test. The data set is made up of 5,957 multiple-choice questions, accompanied by 6,493 commonsense statements about science facts. Since there is a significant overlap between the knowledge in questions and statements, we only use the statements data for simplicity.

We now detail the specific preprocessing method for each source dataset to convert them (i.e., question-answer pairs) into statements.

- If the raw data comes in the statement format (CREAK and OpenbookQA), we obtain the true statements (part of CREAK and all of OpenbookQA) without extra processing.

- If the raw data comes in Yes/No question format (StrategyQA), we leverage a POS-rule-based open-sourced system `question_to_statement`[9] to transform a pair of question and Yes/No answer into a statement.

- If the raw data comes in multiple-choice format (CommonsenseQA and ARC), we utilize a neural model to convert a pair of question and correct choice $(q, a)$ into a statement in a sequence-to-sequence fashion. Concretely, we use the QA-to-statement model checkpoint

---

[9]https://github.com/SunnyWay/question_to_statement

released by [35], which is a BART [22] model finetuned on QA2D [13], a dataset of human-annotated statements for QA pairs.

Converting QA pair to statement is not a difficult task for pretrained seq2seq models. We observe that the generated statements are mostly fluent and faithful to the input. Additionally, we have manually filtered out unnatural examples in the test set. We summarize the basic information of these datasets and provide an example of statement conversion for each dataset in Table 5.

## A.2 Antithesis Mining

**Keyword Masking.**   We use entities and other nouns as the keywords of sentences because as a pilot study, we only consider the relationships between spatio-temporal contexts and nouns and ignore the influence of other part-of-speech categories such as verbs, adjectives, and prepositions. We use the same NER tagger in Section 4.2 to extract entities. We leverage spaCy[10] to extract all the nouns (including proper nouns) from a sentence. We merge the entities and nouns as keywords after removing duplicates. In particular, if a noun and an entity partly overlap (e.g., "**month**" and "a lunar **month**"), we retain the entity when deduplicating.

**Masked Sentence Similarity Matching.** We use the pretrained language model `all-MiniLM-L6-v2`[11] released by SentenceTransformers [40] to obtain high-quality embeddings of keyword-masked sentences. We calculate the cosine similarity to pair highly similar masked sentences. Computing the similarity of all possible sentence pairs requires $\mathcal{O}(n^2)$ time complexity. To accelerate this process, we use the `paraphrase_mining` API of SentenceTransformers [40].

**Rule-based Filtering.**   We devise the following rules to filter invalid sentence pairs based on iterative observation of the data:

- The masked sentence similarity exceeds a certain threshold[12], which indicates parallel sentence structure of antithesis.

- The number of masked keywords (`[UNK]`) of every single sentence should not be more than 5 and less than 2, which controls for a reasonable difficulty of the keyword-to-text generation task.

- Any entity in one sentence does not appear in the other sentence within a pair (including the deformation of entity words, such as singular/plural form, upper/lower case, etc.). This is to avoid both sentences expressing the information of the same entity, while contrastive sentences should describe two opposite things.

- Both of the two sentences contain either GEO entities or TEMP entities (GEO+GEO or TEMP+TEMP), which avoids sentences comparing GEO context to a non-parallel TEMP context (GEO+TEMP).

## B   Dataset Quality Analysis

### B.1   Manual Filtering of the Test Set

To ensure the high quality of the dataset, we manually filter out invalid examples in the test set that are not fluent antitheses or context-dependent. This process is important for the very high human performance shown in Table 3. Table 6 shows the instructions for annotators. We first ask two graduate students with proficiency in English to annotate 100 examples as valid or invalid. They agree with each other (i.e., give the same label) on 88% of examples. The inter-annotator agreement in terms of Cohen's Kappa [12] is 0.76, which indicates substantial agreement [21]. Since the agreement ratio is satisfactory, we ask one of the annotators to complete the rest of the filtering process.

---

[10]`https://spacy.io/models/en#en_core_web_sm`
[11]`https://huggingface.co/sentence-transformers/all-MiniLM-L6-v2`
[12]We set the threshold as 0.8 via manual inspection.

Table 6: Annotator instructions for manual filtering of our dataset.

**Goal**: The objective of our project is to generate high-quality contrastive sentence pairs (antithesis) that incorporate geographical and temporal contexts. These sentence pairs will serve as a means to evaluate machines' commonsense reasoning abilities under different extra-linguistic contexts. We aim to create sentences that require a deep understanding of real-world geographical and temporal entities but can be reasonably confirmed without resorting to external sources like Google or Wikipedia.

**Instructions**: We show a set of keywords and a pair of sentences containing these keywords. Your task is to determine whether this sentence pair satisfies *all* of the following criteria:

1. The sentence pair includes all of the given keywords.
2. Each sentence has at least one entity related to geography or time.
3. Each sentence is fluent and adheres to commonsense knowledge.
4. The two sentences have similar syntactic structures and create a contradiction in semantics.
    - Intuitively, the qualifying two sentences can be connected into a coherent sentence via a conjunction word such as "while", "yet", and "whereas" (e.g., *July is summer in the United States, while July is winter in Australia.*).
5. Swapping any of the geographical or temporal entities between the two sentences could lead to a contradiction with commonsense yet grammatical correctness.
    - For example, for the sentence pair "*July is summer in China. July is winter in Australia.*", if the two geographical entities "China" and "Australia" are swapped, the resulting sentences do not adhere to commonsense anymore: "*July is summer in Australia. July is winter in China.*"

**Examples**:
Keywords: *morning, night, sunrise, sunset*
Sentence 1: "The sky is bright with the sunrise in the early morning."
Sentence 2: "The sky is dark with the sunset in the late night."

Criterion 1: Both sentences include the keywords "morning" and "night."
Criterion 2: Each sentence contains a geographical or temporal entity ("sunrise" and "sunset") related to the context.
Criterion 3: Both sentences are fluent and adhere to commonsense knowledge.
Criterion 4: The sentences have a similar syntactic structure and create a semantic contradiction: "The sky is bright with the sunrise in the early morning, while the sky is dark with the sunset in the late night."
Criterion 5: Swapping the temporal entities "early morning" and "late night" would result in a contradiction: "The sky is bright with the sunrise in the late night, while the sky is dark with the sunset in the early morning."

This example demonstrates how the sentence pairs satisfy the specified criteria of the task.

# C  Experimental Setup

## C.1  Baseline Models

We use HuggingFace [48] implementations of the BART and T5 models. For the decoding method, we adopt the standard beam search with a beam size of 4 for all baseline models. As for checkpoint selection, we save a checkpoint for each epoch and select the checkpoint with the highest `ROUGE-2` on the validation set. Other default hyperparameters are shown in Table 7.

Table 8 shows an example of GPT prompt format, consisting of a fixed instruction ("*Generate a pair of contrastive sentences with the given set of keywords.*") and a few in-context demonstrations ("*Keywords: $c_1, ..., c_k$ \n Sentences: $s_1$ $s_2$*").

## C.2  Evaluation Metrics

We use the standard implementation of BLEU, ROUGE, METEOR, CIDEr, and SPICE in `pycocoevalcap`[13]. As recommended, we adopt the Recall score of BERTSCore[14] and the hash code for evaluation setting is "roberta-large_L17_no-idf_version=0.3.12(hug_trans=4.21.3)-rescaled_fast-tokenizer". In addition, we design and implement `MATCH` to evaluate how well the machines solve

---

[13]https://github.com/salaniz/pycocoevalcap
[14]https://github.com/Tiiiger/bert_score

Table 7: Hyper-parameter settings for all baseline models.

| Parameter | Value |
|---|---|
| epoch | 10 |
| batch size | 32 |
| beam size | 4 |
| max input length | 64 |
| max output length | 128 |
| learning rate | 3e-5 |
| warm-up steps | 500 |

Table 8: An example of InstructGPT prompt format. We only show two in-context demonstrations here for brevity.

Generate a pair of contrastive sentences with the given set of keywords.

Keywords: Kansas, steakhouses, New York City, city, pizzerias
Sentences: Kansas city is known for its steakhouses. New York City is known for its pizzerias.
...
Keywords: seven days, one day, 1,440 minutes, a week
Sentences: There are 1,440 minutes in one day. There are seven days a week.

Keywords: axis, one day, one month, Earth, Moon
Sentences:

the challenge of situated semantic matching (Section 3.2). We now define the keyword matching accuracy MATCH based on mathematical notations introduced in Section 3.1.

$t = (t_1, ..., t_k), t_i \in \{0, 1\}$ indicates that each keyword $c_i$ appears in which sentence in the answer pair $y^{true} = \{s_1^{true}, s_2^{true}\}$. In other words, if $c_i$ *should* appear in $s_1$, then $t_i = 0$; if $c_i$ *should* appear in $s_2$, then $t_i = 1$. $p = (p_1, ..., p_k), p_i \in \{-1, 0, 1\}$ indicates that each keyword $c_i$ appears in which sentence in the output pair $y^{pred} = \{s_1^{pred}, s_2^{pred}\}$. In other words, if $c_i$ *actually* appear in $s_1$, then $p_i = 0$; if $c_i$ *actually* appear in $s_2$, then $p_i = 1$; if $c_i$ does not *actually* appear in both $s_1$ and $s_2$, then $p_i = -1$[15]. We define the matching accutacy of a sentence pair match$(y^{true}, y^{pred})$ as the proportion of correctly matched keywords, which is calculated as $\frac{1}{k} \max(\sum_{i=1}^{k} \mathbb{1}_{t_i=p_i}, \sum_{i=1}^{k} \mathbb{1}_{1-t_i=p_i}) \in [0, 1]$. Here $\mathbb{1}$ is the indicator function. The formula includes both $1 - t$ and $t$ in a symmetric way because the sentence pair is unordered. For the whole test set, we take the average matching accuracy of all examples as MATCH.

We illustrate the computing process of matching accuracy with a simple example. Given [July, China, winter, Australia, summer, July], the answer could be "*July is summer in China. July is winter in Australia.*" So $t = (0, 0, 1, 1, 0, 1)$. If the generated output is "*July is summer in Australia. July is winter in China.*", then $p = (0, 1, 1, 0, 0, 1)$. As a result, the matching accuracy is $4/6 = 0.67$.

As for the implementation, we utilize NLTK[16] to split the output into two sentences. In particular, if there is only one sentence in the output, we append an empty string as the second one; if there are more than two sentences, we only take the former two sentences into consideration. We lemmatize the sentence before determining keyword appearance.

---

[15]By defining $p_i = -1$, MATCH can also reflect the coverage of keywords in the output.
[16]https://www.nltk.org/

