# A Datasheet for SITUATEDGEN

## A.1 Motivation

**1. For what purpose was the dataset created? Was there a specific task in mind? Was there a specific gap that needed to be filled? Please provide a description.**

This dataset aims to probe the commonsense reasoning ability of generative language models through the lens of keyword generation tasks. The task requires machines to compose a pair of contrastive sentences with a given set of keywords containing geographical or temporal entities. Current models lack the ability to correctly reason for the relationship among these entities and thus generate sentences that contradict commonsense knowledge. We hope our dataset could stir more research to fill this gap of generative commonsense reasoning.

**2. Who created the dataset (e.g., which team, research group) and on behalf of which entity (e.g., company, institution, organization)?**

The dataset is created by Yunxiang Zhang and Xiaojun Wan on behalf of the Text Mining and Linguistic Computing Group, Wangxuan Institute of Computer Technology, Peking University. Most part of this paper is done when the first author is at Peking University before moving to University of Michigan.

**3. Who funded the creation of the dataset? If there is an associated grant, please provide the name of the grantor and the grant name and number.**

It is funded by the Text Mining and Linguistic Computing Group, Wangxuan Institute of Computer Technology, Peking University.

## A.2 Composition

**1. What do the instances that comprise the dataset represent (e.g., documents, photos, people, countries)? Are there multiple types of instances (e.g., movies, users, and ratings; people and interactions be tween them; nodes and edges)? Please provide a description.**

The dataset is comprised of pure text data in English, presented in a Jsonline format. The file is composed of a list of instances containing input keywords and targeted outputs.

**2. How many instances are there in total (of each type, if appropriate)?**

Our dataset consists of 8,268 instances. Please refer to Table 2 for detailed information.

**3. Does the dataset contain all possible instances or is it a sample (not necessarily random) of instances from a larger set? If the dataset is a sample, then what is the larger set? Is the sample representative of the larger set (e.g., geographic coverage)? If so, please describe how this representativeness was validated/verified. If it is not representative of the larger set, please describe why not (e.g., to cover a more diverse range of instances, because instances were withheld or unavailable).**

This dataset does not cover all aspects of commonsense knowledge so it does not contain all possible instances. We focus on geographical and temporal commonsense in this work since they provide testbeds for evaluating machines' reasoning ability under different extra-linguistic contexts.

**4. What data does each instance consist of? "Raw" data (e.g., unprocessed text or images) or features? In either case, please provide a description.**

Each instance is a dictionary has the following fields:

- "keywords": a list of keywords as input

- "statement": a string concatenation of two sentences as the target generations

- "ids": the origins of the commonsense statement (from which (train/dev/test) split of which source datasets/corpora) represented in the format of "{src_dataset}::{split}::{id}"

- "statements": a list of the two sentences in "statement" field.

**5. Is there a label or target associated with each instance? If so, please provide a description.**

Yes. It is represented as the "statement" filed in each instance.

**6. Is any information missing from individual instances? If so, please provide a description, explaining why this information is missing (e.g., because it was unavailable). This does not include intentionally removed information, but might include, e.g., redacted text.**

No. All instances are complete.

**7. Are relationships between individual instances made explicit (e.g., users' movie ratings, social network links)? If so, please describe how these relationships are made explicit.**

Individual instances are independent of each other. The train/dev/test splits do not overlap in any single sentence.

**8. Are there recommended data splits (e.g., training, development/validation, testing)? If so, please provide a description of these splits, explaining the rationale behind them.**

Yes, see Tabel 2 for details. The splitting process makes sure that the train/dev/test splits do not overlap in any single sentence. See Appendix D.3 for details.

**9. Are there any errors, sources of noise, or redundancies in the dataset? If so, please provide a description.**

There is noise in the train and dev set. We manually filter out unqualified examples in the test set. See more analysis in Section 5.1 and Appendix E.2.

**10. Is the dataset self-contained, or does it link to or otherwise rely on external resources (e.g., websites, tweets, other datasets)? If it links to or relies on external resources, a) are there guarantees that they will exist, and remain constant, over time; b) are there official archival versions of the complete dataset (i.e., including the external resources as they existed at the time the dataset was created); c) are there any restrictions (e.g., licenses, fees) associated with any of the external resources that might apply to a dataset consumer? Please provide descriptions of all external resources and any restrictions associated with them, as well as links or other access points, as appropriate.**

The SITUATEDGEN dataset is self-contained and we welcome practitioners to consider additional knowledge sources.

**11. Does the dataset contain data that might be considered confidential (e.g., data that is protected by legal privilege or by doctor–patient confidentiality, data that includes the content of individuals' non-public communications)? If so, please provide a description.**

No.

**12. Does the dataset contain data that, if viewed directly, might be offensive, insulting, threatening, or might otherwise cause anxiety? If so, please describe why.**

No.

### A.3 Collection Process

**1. How was the data associated with each instance acquired? Was the data directly observable (e.g., raw text, movie ratings), reported by subjects (e.g., survey responses), or indirectly inferred/derived from other data (e.g., part-of-speech tags, model-based guesses for age or language)? If the data was reported by subjects or indirectly inferred/derived from other data, was the data validated/verified? If so, please describe how.**

The data is sourced from several commonsense related datasets and corpora. We design an automatic pipeline to convert and filter data into our desired format.

**2. What mechanisms or procedures were used to collect the data (e.g., hardware apparatuses or sensors, manual human curation, software programs, software APIs)? How were these mechanisms or procedures validated?**

We first convert instances from other datasets as commonsense statements. Then we match these statements into pairs and extract keywords from them. We further manually filter out invalid examples in the test set.

**3. Who was involved in the data collection process (e.g., students, crowdworkers, contractors) and how were they compensated (e.g., how much were crowdworkers paid)?**

We hired crowdworkers and compensated them with 0.1 yuan for each entry they checked, which is higher than the statutory minimum wage.

**4. Over what timeframe was the data collected? Does this timeframe match the creation timeframe of the data associated with the instances (e.g., recent crawl of old news articles)? If not, please describe the time frame in which the data associated with the instances was created.**

Our dataset was built in 2022 while the original source data is published between 2018-2021. Usually, commonsense statements are not changing over time.

**5. Were any ethical review processes conducted (e.g., by an institutional review board)? If so, please provide a description of these review processes, including the outcomes, as well as a link or other access point to any supporting documentation.**

No.

### A.4 Preprocessing/cleaning/labeling

**1. Was any preprocessing/cleaning/labeling of the data done (e.g., discretization or bucketing, tokenization, part-of-speech tagging, SIFT feature extraction, removal of instances, processing of missing values)? If so, please provide a description. If not, you may skip the remaining questions in this section.**

Yes. We use templated-based and neural-based models to convert and filter the source data into our desired format. See details in Appendix D.

**2. Was the "raw" data saved in addition to the preprocessed/cleaned/labeled data (e.g., to support unanticipated future uses)? If so, please provide a link or other access point to the "raw" data.** Yes. The raw data is available on the corresponding dataset websites (CREAK – `https://github.com/yasumasaonoe/creak`, OpenbookQA – `https://allenai.org/data/open-book-qa`, StrategyQA – `https://allenai.org/data/strategyqa`, CommonsenseQA – `https://www.tau-nlp.sites.tau.ac.il/commonsenseqa`, ARC – `https://allenai.org/data/arc`).

**3. Is the software that was used to preprocess/clean/label the data available? If so, please provide a link or other access point.**

Yes. Please see `https://github.com/yunx-z/situated_gen`.

### A.5 Uses

**1. Has the dataset been used for any tasks already? If so, please provide a description.**

Not yet.

**2. Is there a repository that links to any or all papers or systems that use the dataset? If so, please provide a link or other access point.**

There has not been such a paper or system yet.

**3. What (other) tasks could the dataset be used for?**

It can be used to develop better language models for commonsense reasoning. It can be used to evaluate language models, especially their understanding of commonsense knowledge. It could potentially benefit many downstream applications such as document summarization [44], story writing [51] and dialogue response generation [31].

**4. Is there anything about the composition of the dataset or the way it was collected and preprocessed/cleaned/labeled that might impact future uses? For example, is there anything that a dataset consumer might need to know to avoid uses that could result in unfair treatment of individuals or groups (e.g., stereotyping, quality of service issues) or other risks or harms (e.g., legal risks, financial harms)? If so, please provide a description. Is there anything a dataset consumer could do to mitigate these risks or harms?**

The dataset has very low risks of containing sentences with toxicity and offensiveness. Since our data is sourced from existing datasets, we may inherit geographical biases [16] that result in an uneven distribution of commonsense knowledge about western and non-western regions. The commonsense statements may not sound familiar to people who live in locations that are poorly represented in the source datasets. Therefore, models developed on our dataset may preserve biases learned from the annotators of the source datasets. We note that pretrained language models may also inherit the bias in the massive pretraining data. It is important that interested parties carefully address those biases before deploying the model to real-world settings.

**5. Are there tasks for which the dataset should not be used? If so, please provide a description.**

The dataset can only be used for research purposes.

### A.6 Distribution

**1. Will the dataset be distributed to third parties outside of the entity (e.g., company, institution, organization) on behalf of which the dataset was created? If so, please provide a description.**

The dataset is already publicly available.

**2. How will the dataset will be distributed (e.g., tarball on website, API, GitHub)? Does the dataset have a digital object identifier (DOI)?**

The dataset is available at `https://github.com/yunx-z/situated_gen`.

**3. When will the dataset be distributed?**

It has already been distributed.

**4. Will the dataset be distributed under a copyright or other intellectual property (IP) license, and/or under applicable terms of use(ToU)? If so, please describe this license and/or ToU, and provide a link or other access point to, or otherwise reproduce, any relevant licensing terms or ToU, as well as any fees associated with these restrictions.**

This dataset is licensed under the Creative Commons Attribution-NonCommercial-ShareAlike 4.0 International License (CC BY-NC-SA 4.0). The full text of the license can be accessed at the following link: `https://creativecommons.org/licenses/by-nc-sa/4.0/`.

**5. Have any third parties imposed IP-based or other restrictions on the data associated with the instances? If so, please describe these restrictions, and provide a link or other access point to, or otherwise reproduce, any relevant licensing terms, as well as any fees associated with these restrictions.**

No.

**6. Do any export controls or other regulatory restrictions apply to the dataset or to individual instances? If so, please describe these restrictions, and provide a link or other access point to, or otherwise reproduce, any supporting documentation.**

No.

## A.7 Maintenance

**1. Who will be supporting/hosting/maintaining the dataset?**

The first author, Yunxiang Zhang, is hosting and maintaining the dataset.

**2. How can the owner/curator/manager of the dataset be contacted (e.g., email address)?**

Email: yunxiang@umich.edu

**3. Is there an erratum? If so, please provide a link or other access point.**

No.

**4. Will the dataset be updated (e.g., to correct labeling errors, add new instances, delete instances)? If so, please describe how often, by whom, and how updates will be communicated to dataset consumers (e.g., mailing list, GitHub)?**

We are interested to collect more data using our automatic pipelines and conduct manual filtering as future work. We also welcome interested parties to point out errors in the dataset via contact email or github issues so we could correct them. If there is a plan for systematic updates, we will announce it at the earliest opportunity.

**5. If others want to extend/augment/build on/contribute to the dataset, is there a mechanism for them to do so? If so, please provide a description. Will these contributions be validated/verified? If so, please describe how. If not, why not? Is there a process for communicating/distributing these contributions to dataset consumers? If so, please provide a description.**

People can use this repository following the licenses and cite our paper.