# OpenReview forum: "SituatedGen: Incorporating Geographical and Temporal Contexts into Generative Commonsense Reasoning"
_NeurIPS.cc/2023/Track/Datasets_and_Benchmarks — NeurIPS 2023 Datasets and Benchmarks Poster_

### Official Review · Reviewer_jj95 · 2023-07-18
**The paper brings a new dataset on antithesis related pairs of sentences semi-automatically from existing datasets for geographical and temporal scenarios.**

**Rating:** 7
**Confidence:** 3

**Strengths:**

1. The paper presents a method to automatically collect contrastive sentences at scale with minimal supervision. The technique of entity-masked sentence similarity is a simple and elegant way to derive such sentences with a reasonable reliability – useful to mine antithesis, although it relies on reliably identifying the entities to be masked.
2. Within the bounds of the goal to study geo and temporal contrastive commonsense, the task setup is very clearly defined especially in the generation case. Specifically Section 3.1 is well thought and laid out.



**Additional Feedback:**

Minor grammar corrections:
Line 118: “yet grammatical correctness” -> “yet remaining grammatically correct” (or other corrections to complete the sentence).



**Clarity:**

The paper is very well written – special mention to Section 3.1 is very well thought of and the space is clearly defined.

**Correctness:**

The paper mentions a large scale dataset. The algorithm to mine these antithesis sentence pairs is semi-automatic and can be scaled well. However, the number of sentences collected are not too high. Seems like 87% of these are real context-dependent antithesis which further shrinks the dataset size, which are all retained in the data to ensure a reasonable size of the training data.

[+] The memorization aspect is well-thought of during dataset construction and the authors made sure that there is not even a single statement overlapping between the train, val and test sets. This is extended to syntactic overlap as well.


**Documentation:**

The dataset is well-documented and can be used with ease.

**Ethics:**

The methodology to construct the dataset does not include any new biases. However, if the relying base datasets have any kinds of biases, they are directly reflected in this dataset as well.

**Limitations:**

The way to evaluate each of the sentence independently and relative to one another brings forth more challenges in evaluation. The evaluation metrics capture the coverage, correctness, consistency, and the content of the sentences. The specific aspects of contrast between the sentences that are tailored for this dataset are not being evaluated by the metrics.


**Opportunities For Improvement:**

The generative pretrained models decode are trained to consistently generate about a topic. The two sentences in the antithesis are specifically unrelated on purpose. From the examples shown and as mentioned in the paper, it seems like the models miss mentioning the keywords. It is plausible that since the keywords are regarding unrelated topics, and the decoder of the models are trying to be consistent about a topic, the keywords are missed in generation. Providing this as additional information in the form of a prefix might be useful especially for instruction-tuned models.


**Relation To Prior Work:**

The paper clearly describes the keyword based generation datasets and the tasks and how this dataset differs from them.


**Summary And Contributions:**

While most of the commonsense reasoning work is focused on general scenarios in task setup for understanding, this work explores it in a more focused area of temporal and geographical cases in generation setup. The task is to generate contrastive sentences for the same keywords where only certain combination is plausible and collected ~8k contrastive sentences to this end. To automatically collect this dataset, a simple method to mask the entities in the sentence to compute similarity is used. An additional step of manual curation is done to ensure the quality of the test data. Datasets from different origins for commonsense in the format of question answering are normalized to the form of statements that resulted in ~35k statements. Manual filtering is applied for the test set and the authors made sure that there is no overlap between the train, val and test sets. The proposed dataset is benchmarked with seven models trained with the permutation of 2 sentences in each pair where InstructGPT shows good results on automatic metrics.

The main contributions of the paper are:
1. In contrast to the majority of the work in commonsense reasoning setup as discriminative tasks, this work follows the generation path (not new but minority of the work like CommonGen).
2. Instead of the general scenarios, this work focuses on specific contexts for geographical and temporal commonsense reasoning enabling us to delve deeper into the edge cases.
3. They construct a new dataset of ~8k sentence pairs along with keywords to study this task further and release the dataset publicly.
4. They study the performance of the existing models on the task and show the

---

> ### Author Response · Authors · 2023-08-13
> **Author Response**
>
> Re: Opportunities For Improvement
>
> Thank you for your suggestion. We note that we have already specified the keyword constraint in the instruction/prompt for InstructGPT (“Generate a pair of contrastive sentences with the given set of keywords.” See Table 8 in Appendix F). So the LLM should be aware that it is required to include all keywords in the generation results. We will explore the effect of different instructions in the future.
>
>
> Re: Limitation
>
> To evaluate the aspect of contrast between the sentences that you mentioned, one can try to compute the similarity between the entity-masked sentences. This is similar to how we mine contrastive sentences during dataset collection (Fig 2 Section 4.3). We agree that automatic evaluation is a challenging problem for our task and there is still room for improvement.

---

> > ### Comment · Reviewer_jj95 · 2023-08-30
> >
> > The idea about using entity-masking to mine sentences is a decent fix to circumvent human evaluation. While a similar technique is used for mining sentences, it would be good to understand the effectiveness of the method to see the aspects. Seeing that we are on the same page, I have no further questions. The paper is a valuable read for target audience.

---

### Official Review · Reviewer_rfw9 · 2023-07-20
**Review of SituatedGen**

**Rating:** 7
**Confidence:** 3

**Strengths:**

The authors effectively argued the importance of considering external context (i.e., geo and temporal) in understanding commonsense relationship. Considering commonsense relationship can be brittle to small textual manipulation, it is useful for related communities to provide contrastive sentence pairs. Contrastive pairs will help enhance robustness of model performance in many situations. Furthermore, the authors sophisticatedly split train/validation/test set so that overlapping lexical information cannot intervene model performance.

**Additional Feedback:**

N/A

**Clarity:**

This paper is very well written with appropriate examples supporting their arguments.

**Correctness:**

The dataset is constructed in automatic pipeline with manual filtering. The overall process is sound and the authors guided detailed information (e.g., used software, algorithm) in the supplementary material.

**Documentation:**

The overall process is clearly described.

**Limitations:**

In Limitation section, the authors specified potential errors transferred from previous works.

**Opportunities For Improvement:**

Due to its structural constraints (e.g., maintaining antithesis structure between contrastive sentences), it seems hard to construct multi-references given set of keywords, which is different from CommonGen that crowdsourced multiple plausible references of the combination of keywords. Furthermore, it would be great if the authors investigated the robustness to small manipulation changing commonsense relationship, not only employing previous metrics.

**Relation To Prior Work:**

In Related Work section, the authors discussed the difference with previous works. It can be organized as follows:
- Focused on constructing contrastive sentence pairs instead of single sentence
- Expanded the range of commonsense reasoning by incorporating external context

**Summary And Contributions:**

This paper proposed SituatedGen, a generative commonsense benchmark conditioned by geographical and temporal context. The authors collected existing commonsense benchmarks and reformulated them as contrastive sentence pairs using automatic pipeline. They described dataset collection pipeline in detail, and showed the baseline performance using representative language models.

---

> ### Author Response · Authors · 2023-08-13
> **Author Response**
>
> Re: Opportunities For Improvement:
>
> Different from CommonGen where one could devise multiple plausible scenarios for a single set of keywords, the correct geographical or temporal contexts for each keyword set in our SituatedGen are often constrained to a particular case. We thus argue that multi-references do not have a crucial impact on the soundness of our evaluation. Yet we do agree that it would be beneficial to evaluate the robustness of models to small input perturbations, which we leave as future work to explore.

---

> > ### Comment · Reviewer_rfw9 · 2023-08-29
> >
> > I hope further works handle relevant interesting questions, so mentioning potential research agenda in the paper would help other researchers to engage in. I have no further questions.

---

### Official Review · Reviewer_63Tn · 2023-07-21
**Commonsense Reasoning Dataset Review**

**Rating:** 6
**Confidence:** 3
**Clarity:** The paper is well-written and easy to…

**Strengths:**

1. The dataset is of high quality and seems meaningful in commonsense reasoning research.
2. Extensive experiments demonstrate the difficulty of the task and the limitation of existing generative LMs.
3. The paper is well written. Rich details are provided in the appendix.

**Additional Feedback:**

In Table 3, GEO and TEMP in the model column may be confusing. Some explanations in the caption will make it more clear.

**Correctness:**

The paper overall seems correct, with a clear explanation of the dataset construction and the evaluation methods.

**Documentation:**

The paper contains details on how the data was collected and where it is maintained. Sufficient details to support reproducibility is included in the paper.

**Ethics:**

There seem to be no ethical concerns.

**Limitations:**

The authors discussed the limitations in the appendix.

**Opportunities For Improvement:**

1. The paper concludes that LMs lag far behind humans in generating commonsensical sentences. However, the LMs are mostly not that large. The largest one, InstructGPT, is evaluated in a simple in-context setting. I believe conducting experiments that include reasoning steps (eg, chain-of-thought) or simple instructions in the prompt is necessary to make a better comparison. Just giving LLM a buffer zone before reaching the reasoning result might boost the performance.

2. How well can models explicitly classify the keywords into two subgroups? In the experiments, this process seems to be implicit. If the reasoning is broken into two steps: explicitly classify the keywords into two subgroups and then generate the two sentences, will the LMs do better? With the process implicit, LMs' reasoning ability may not be fully explored.

3. Line 149 - 152 is a little confusing, why this can prevent models from learning shortcuts to group keywords based on trivial syntactic?

**Relation To Prior Work:**

The paper discusses related work well.

**Summary And Contributions:**

The paper curates a new commonsense reasoning dataset focusing on geographical and temporal contexts. Moreover, the paper proposes a new task that requires generating a pair of contrastive sentences given some geographical and temporal entities. The authors evaluate several generative language models on the dataset.

Contributions:
1. Curate a new commonsense reasoning dataset that incorporates geographical and temporal contexts
2. Propose a new reasoning task
3. benchmark the performance of several language models on the dataset

---

> ### Author Response · Authors · 2023-08-13
> **Author Response**
>
> Re: Opportunities For Improvement 1.
>
> We agree that it would be beneficial to include a more comprehensive evaluation of recent LLMs with advanced prompting techniques. We will try to report new results in the camera ready if our budget is sufficient to run experiments with these API-based LLMs.
>
> Re: Opportunities For Improvement 2.
>
> It would be interesting to devise models that can decompose the reasoning into two stages: keyword grouping and sentence generation. Due to the space limitation of this paper, we leave it as future work to study.
>
> Re: Opportunities For Improvement 3.
>
> Thank you for the clarification question and please find the explanation below. If the two sentences have different syntactic structures (e.g. “X lives in Y” and “Z eats W”), then the model could simply put a city name in Y and a food name in W for keyword grouping and ignore the commonsense connection with X/Z. This type of shortcut reduces the task difficulty. This also motivates our dataset collection algorithm in Section 4.3 (also Figure 2) --- after we mask the entities of the sentence, the rest is basically the syntactic structure of the sentence and we form sentence pairs with similar syntactic structure.

---

> > ### Comment · Reviewer_63Tn · 2023-08-25
> >
> > Thank you for answering my questions and providing clarifications. I have no further questions.

---

### Author Response · Authors · 2023-08-13
**To All Reviewers**

We thank all the reviewers for their thoughtful reviews and constructive feedback. We are glad you recognize the novelty of our proposed reasoning task, our dataset's quality, and the evaluation's soundness. We prepare separate responses for each reviewer. Please let us know if there are any follow-up questions or comments.

---

### Decision · Program_Chairs · 2023-09-22

**Decision:**

Accept (Poster)

**Comment:**

Based on reviewer discussions, the Created dataset has merits.